# A Supervised LightGBM-Based Approach to the GSK.ai CausalBench Challenge (ICLR 2023)
# Team Guanlab Report Submissions

## Abstract

In this challenge, we transformed the task of detecting gene pairs with causal relationships into a supervised learning problem. We constructed a dataset for all gene pairs, with initial labels determined by gene expression correlations. A LightGBM model was trained and applied to the same data for prediction. The top 1001 pairs with the highest prediction scores were selected. In local experiments, this solution achieved a 0.3779 AUC score in the RPE1 data and a 0.3265 score in the K562 data.

## 1 Notations

In addition to standard notations, we defined several custom notations listed below to describe the method more efficiently.

| | |
|---|---|
| $\langle \boldsymbol{g}_i, \boldsymbol{g}_j \rangle$ | A directed gene pair from $\boldsymbol{g}_i$ to $\boldsymbol{g}_j$ |
| $\boldsymbol{M}_{\boldsymbol{g}_i, \boldsymbol{g}_j}$ | Select the rows for $\boldsymbol{g}_i$ and the columns for $\boldsymbol{g}_j$ from the expression matrix $\boldsymbol{M}$ |
| $\overline{\mu_{\boldsymbol{M},0}}$ | The column-wise mean value of the expression matrix |
| $\overline{\sigma_{\boldsymbol{M},0}}$ | The column-wise standard deviation of the expression matrix |

## 2 Methods

### 2.1 Calculate the correlations

We calculated correlations for all possible gene pairs $\langle \boldsymbol{g}_i, \boldsymbol{g}_j \rangle$, where $\boldsymbol{g}_i$ and $\boldsymbol{g}_j$ belonged to the columns of the expression matrix $M_{k \times l}$ and $i \neq j$. The input expression data were the concatenation of the interventional data ($\boldsymbol{M}_{\boldsymbol{g}_i, \boldsymbol{g}_i}$, $\boldsymbol{M}_{\boldsymbol{g}_i, \boldsymbol{g}_j}$) and the samples from the observational data ($\boldsymbol{M}_{non-targeting, \boldsymbol{g}_i}$, $\boldsymbol{M}_{non-targeting, \boldsymbol{g}_j}$). The observational data samples had the same lengths as the interventional data. If $\boldsymbol{g}_i$ related cells were not present in the expression matrix due to partial selection, the input data would be $\boldsymbol{M}_{non-targeting, \boldsymbol{g}_i}$ and $\boldsymbol{M}_{non-targeting, \boldsymbol{g}_j}$. The resulting correlation matrix was asymmetric and had the shape of $(l, l)$.

### 2.2 Construct the dataset

The initial labels of gene pairs were determined using a correlation threshold $T$. Pairs with correlation scores higher than 0.1 were labeled as positive samples. To generate the features, we first normalized the expression matrix using $(\boldsymbol{M} - \overline{\mu_{\boldsymbol{M},0}})/\overline{\sigma_{\boldsymbol{M},0}}$. For each gene pair $\langle \boldsymbol{g}_i, \boldsymbol{g}_j \rangle$, we extracted four features from the matrix: $\overline{\boldsymbol{M}_{non-targeting, \boldsymbol{g}_i}}$, $\overline{\boldsymbol{M}_{non-targeting, \boldsymbol{g}_j}}$ (average observational expression of $\boldsymbol{g}_i$ and $\boldsymbol{g}_j$), and $\overline{\boldsymbol{M}_{\boldsymbol{g}_i, \boldsymbol{g}_i}}$, $\overline{\boldsymbol{M}_{\boldsymbol{g}_i, \boldsymbol{g}_j}}$ (average intervened expression by $\boldsymbol{g}_i$). If $\boldsymbol{g}_i$ related

Table 1: LightGBM hyper-parameters

| Parameter | Value |
|---|---|
| boosting_type | gbdt |
| objective | binary |
| metric | binary_logloss |
| num_leaves | 5 |
| max_depth | 2 |
| min_data_in_leaf | 5 |
| learning_rate | 0.05 |
| min_gain_to_split | 0.01 |
| num_iterations | 1000 |

cells were missing in the expression matrix, the last two features would be 0 and NaN. The output dataset would have $l \times (l-1)$ rows and 5 columns.

### 2.3 TRAIN THE MODEL AND PREDICT

The LightGBM model was set up using the hyperparameters listed in Table 1 and trained on the entire dataset. Predictions were from applying the model to the same data used for training. We selected the top 1001 gene pairs with the highest prediction scores as our final outputs.

## 3 EXPERIMENTS

To determine the details of training parameters, including methods for initializing positive samples ($K$ and $T$), the number of negative samples ($R$), the number of output gene pairs ($N$), normalization methods, and ensembles, we established two stages of experiments on partial intervention data with one partial seed and five partial seeds.

$K$ and $T$ were parameters for selecting positive samples. We labeled the top $K$ correlated pairs or those with scores higher than $T$ as positive samples. In some experiments, we randomly selected $K \times R$ negative samples and trained the model alongside the positive ones. We also attempted to train multiple models for the ensemble by selecting different negative samples. The ensemble prediction scores were the averages from these models.

Evaluation scores were AUCs. In the first stage, we observed that top-performing methods might have controversial results in K562 and RPE1 and close scores (Table 2). These methods were selected for the second stage evaluation, where we determined the final submission (Table 3).

## 4 DISCUSSION

In summary, we developed a supervised algorithm to solve the unsupervised gene causality prediction problem. Our experiments demonstrated the model's ability to learn the relationships that determined causalities from the expression data and correct false positive and false negative samples from initial labels. The model might benefit from the uncertainty of the initial labels, as including more moderately correlated pairs as positive samples could improve performance. We observed about 0.1 to 0.2 AUC score improvements compared to GRNBoost and DCDI baseline models, in which we also selected the top 1000 pairs as outputs.

We attempted to incorporate the correlation matrix into the baseline algorithms. Since GRNBoost had the highest Wasserstein scores when only considering observational data, we first selected 20,000 candidates with the highest feature importance scores from the model trained on observational data and chose 1000 based on correlation scores. However, this approach failed to surpass direct correlation usage. As the number of candidates in the first selection increased, performance approached the correlation results, suggesting that the GRNBoost model might not provide information beyond correlations.

Table 2: Performances of 1 partial seed data

| K or T | N | R | Normalize | Ensemble | K562 | RPE1 |
|---|---|---|---|---|---|---|
| Top 1000 absolute correlation (baseline) | | | | | 0.2890 | 0.3397 |
| 500 | 1000 | 2 | / | / | 0.1861 | 0.3040 |
| 2000 | 1000 | 2 | / | / | 0.2393 | 0.3352 |
| 2000 | 1000 | 3 | / | True | 0.2561 | 0.3608 |
| 2000 | 2000 | 3 | / | / | 0.2278 | 0.2767 |
| 5000 | 1000 | 2 | / | / | 0.2524 | 0.3552 |
| 5000 | 1000 | 2 | / | True | 0.2614 | 0.3541 |
| 5000 | 1000 | 3 | / | True | 0.2635 | 0.3598 |
| 7000 | 1000 | 3 | / | / | 0.2684 | 0.3608 |
| 7000 | 1000 | AllNeg | / | / | 0.2826 | 0.3846 |
| 7000 | 1000 | AllNeg | normalize | / | 0.3023 | 0.3744 |
| 7000 | 1000 | AllNeg | quantile | / | 0.2843 | 0.3768 |
| 0.1 | 1000 | AllNeg | normalize | / | 0.3148 | / |
| 0.2 | 1000 | AllNeg | normalize | / | 0.3072 | / |

Table 3: Performances of 5 partial seeds data

| K or T | N | R | Normalize | Ensemble | K562 | RPE1 |
|---|---|---|---|---|---|---|
| Top 1000 absolute correlation (baseline) | | | | | 0.2922 | 0.3255 |
| 5000 | 1000 | AllNeg | / | / | 0.2930 | 0.3672 |
| 5000 | 1000 | AllNeg | normalize | / | 0.3062 | 0.3655 |
| 5000 | 1000 | AllNeg | quantile | / | 0.2992 | 0.3632 |
| 7000 | 1000 | AllNeg | / | / | 0.2944 | 0.3659 |
| **0.1** | **1000** | **AllNeg** | **normalize** | **/** | **0.3265** | **0.3780** |
| 0.2 | 1000 | AllNeg | normalize | / | 0.3138 | 0.3614 |

For the DCDI algorithms, we tried replacing the initial adjacency matrix and the Gumbel adjacency matrix with knowledge from the correlation matrix. The improvement over the baseline was nearly 0.1 but still worse than directly using the correlation matrix. Additionally, the algorithm seemed vulnerable to node numbers. We were unable to increase gene numbers for each partition as the program reported overflow issues.

AUTHOR CONTRIBUTIONS

YG and KD design, implement the algorithm; write, and proofread the report.

