# OpenReview forum: "A Supervised LightGBM-Based Approach to the GSK.ai CausalBench Challenge (ICLR 2023)"
_GSK.ai/2023/CBC_

### Official Review · Reviewer_tYuC · 2023-04-28
**LightGBM to predict binarized correlations from mean expression**

**Rating:** 6
**Confidence:** 4

**Review:**

This is a difficult paper to review without having yet evaluated their performance on the challenge metrics because many of the cons would become pros if they do well.  The reason is that the method itself is very reductionist in that the pairwise correlations are thresholded, and thus binarized, and then predicted from just four features, all of which being different averages of the gene's expression values.  So if they do well, then that's pretty damning of all other methods and baselines since they do so with such a simple approach.  However on the flip side if they do poorly then this is perhaps an oversimplification of the problem.

Pros
1.  Simple model, simple inputs and outputs
2.  Decent amount of hyper parameter tuning and search

Cons
1. Maybe too simple
2. Their proxy for a gene to gene 'edge' is a binarized correlation, which seems like it would waste a lot of signal.  Especially since they could just as well fit a model that predicts the correlation itself.  And it would seem that there could be many biologically relevant gene relationships that are missed by just caring about whether their correlation is greater than 0.1 for example.
2. Lack of clarity in exposition.  Model metrics reported are specific to their reformulation of the task.  The tables describing the hyperparamter searches are not very clear.  Hard to tell wha the ratio of positive/negative samples is per row and thus how to judge the AUC metrics.  Unclear to me what "N" is in those tables